# Experimental Study on Precipitation Behavior of Spinels in Stainless Steel-Making Slag under Heating Treatment

**Jianli Li [1,2,3,]\*** **, Qiqiang Mou [1], Qiang Zeng [1] and Yue Yu [2,3]**

[1]  The State Key Laboratory of Refractories and Metallurgy, Wuhan University of Science and Technology, Wuhan 430081, China
[2]  Hubei Provincial Key Laboratory for New Processes of Ironmaking and Steelmaking, Wuhan University of Science and Technology, Wuhan 430081, China
[3]  Key Laboratory for Ferrous Metallurgy and Resources Utilization of Ministry of Education, Wuhan University of Science and Technology, Wuhan 430081, China
\*  Correspondence: jli@wust.edu.cn; Tel.: +86-134-1956-9956

**Abstract:** The stability of chromium in stainless steel slag has a positive correlation with spinel particle size and a negative correlation with the calcium content of the spinel. The effect of heating time on the precipitation of spinel crystals in the $CaO\text{-}SiO_2\text{-}MgO\text{-}Al_2O_3\text{-}Cr_2O_3\text{-}FeO$ system was investigated in the laboratory. Scanning electron microscopy with energy-dispersive and X-ray diffraction were adopted to observe the microstructure, test the chemical composition, and determine the mineral phases of synthetic slags, and FactSage7.1 was applied to calculate the crystallization process of the molten slag. The results showed that the particle size of the spinel crystals increased from 9.42 to 10.73 μm, the calcium content in the spinel crystals decreased from 1.38 at% to 0.78 at%, and the content of chromium in the spinel crystal increased from 16.55 at% to 22.78 at% with an increase in the heating time from 0 min to 120 min at 1450 °C. Furthermore, the species of spinel minerals remained constant. Therefore, an extension in the heating time is beneficial for improving the stability of chromium in stainless steel slag.

**Keywords:** stainless steel slag; heating time; $Cr_2O_3$; spinel; crystal size

## 1. Introduction

Stainless steel slag (SS slag) is a solid waste discharged from the production of stainless steel, and includes electric arc furnace (EAF) slag and argon-oxygen decarbonization furnace (AOD) slag [1]. The amount of SS slag is growing with the increase in demand for stainless steel. However, the toxicity of SS slag is extremely high due to its $Cr^{6+}$ content [2,3], and untreated SS slag threatens the ecological environment and human health [4–6]. Thus, it improving the chromium stability in SS slag is urgent. Chromium spinels not only prevent the leaching of chromium, but can also inhibit the oxidation of $Cr^{3+}$ [7], which is regarded as an ideal mineral phase for stabilizing chromium.

Thus far, much work has been focused on the relationship between spinel precipitation and chromium stability in order to prove that the stability of chromium can be enhanced by increasing spinel precipitation. Zeng et al. [8] found that when the size of the spinel crystals in the slag was increased from 5.77 μm to 8.40 μm, the concentration of the Cr (VI) leaching decreased from 0.1434 mg/L to 0.0021 mg/L. Zhao et al. [9] reported the same phenomenon. When the size of the was spinel increased from 6.0 μm to 17.3 μm, the concentration of the Cr (VI) leaching decreased from 1.24 mg/L to <0.01 mg/L. Therefore, the size of the spinel crystal plays an important role in the stability of chromium in steelmaking slags containing chromium. Regarding the size of the spinel crystal, Wang et al. [10]

adopted the addition of $B_2O_3$ to promote an increase in the spinel size from 7.87 µm to 12.72 µm and the concentration of chromium from 29.69% to 81.90%. Zhang et al. [11] reported that when the content of $Al_2O_3$ was increased from 0 to 15%, the spinel size increased from <20 µm to 23.50 µm. However, most of these investigations were mainly focused on the effect of additions to the spinel characteristics, and there have been few studies on the influence of heat treatment on the precipitation behavior of spinels in SS slag.

In this paper, the effect of heating time on the precipitation of spinels in $Ca-SiO_2-MgO-Al_2O_3-Cr_2O_3-FeO$ was investigated based on characterizations of synthetic samples through scanning electron microscopy equipped with an energy dispersive spectrometer (SEM-EDS, NanoSEM400, FEI, Hillsboro, OR, USA), x-ray diffraction (XRD, X Pert Pro MPD, PANalytica, Almelo, The Netherlands), Image-Pro Plus 6.0 (IPP, Media Cybernetics, MD, USA), and FactSage 7.1 (GTT-Technologies, Aachen, Germany).

## 2. Experimental Process

The slag system was $CaO-SiO_2-MgO-Al_2O_3-Cr_2O_3-FeO$ and the component of the slag sample is shown in Table 1. The raw materials included CaO, $SiO_2$, MgO, $Al_2O_3$, $Cr_2O_3$, and $FeC_2O_4 \cdot 2H_2O$ (Sinopharm Chemical Reagent Co., Ltd., Shanghai, China). First, the raw materials were weighed based on Table 1 and then mixed. 0.10 wt% $H_3BO_3$ (Sinopharm Chemical Reagent Co., Ltd., Shanghai, China) was added into the mixtures to prevent the pulverization of the synthetic samples. Second, mixtures weighing 200 g were put into a molybdenum crucible in the carbon-tube furnace. The furnace was warmed to 1600 °C at a heating rate of 10 °C/min under a nitrogen atmosphere and held for 30 min. Then, the temperature was decreased to 1450 °C at a rate of 20 °C/min, and the slag samples were taken out from the furnace and quenched with water to obtain synthetic slag samples at different sampling times, as shown in Figure 1. The sampling times were 0, 5, 10, 20, 30, 40, 60, 80, 100, and 120 min. Finally, the slag samples were ground, polished, and sprayed with gold powder.

**Table 1.** The chemical composition of stainless-steel slag, g.

| CaO | SiO$_2$ | MgO | Al$_2$O$_3$ | Cr$_2$O$_3$ | FeO | CaO%/SiO$_2$% |
|------|------|------|------|------|------|------|
| 46.67 | 33.33 | 8.00 | 6.00 | 6.00 | 8.00 | 1.40 |

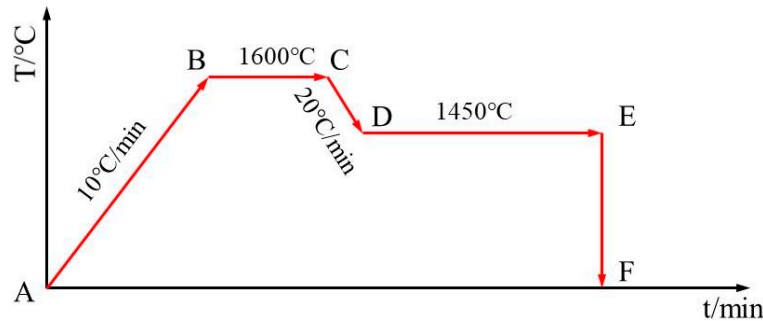

**Figure 1.** The heating procedure adopted in the experiments.

The characterization of the samples was conducted with SEM-EDS and XRD. The size of the spinel crystals was measured with Image-Pro Plus 6.0 (IPP). According to the Scheil-Gulliver equation, FactSage7.1 was applied to simulate the phase transformation and precipitation of the spinel phase during solidification of the molten slag. The specific setting conditions are shown as follows.

Database: FactPS, FToxide, FSstel;

Compound setting: idea gas, pure solid;

Solution phase setting: FToxid-SLAGA, FToxid-SPINA, FToxid-MeO_A, FToxid-bC2SA, FToxid-aC2SA, FToxid-Mel_A.

The FToxid-SLAGA was set as the target phase of Scheil-Gulliver cooling. The setting temperature was 2000 °C and the solidification step was 10 °C. The calculation process was terminated automatically when the target phase completely disappeared. The results were exported as pictures and edited with FactSage7.1.

## 3. Experimental Results

### 3.1. Mineral Phase of CaO-SiO$_2$-MgO-Al$_2$O$_3$-Cr$_2$O$_3$-FeO System

The XRD diffraction spectrum of the slag sample at different sampling times is shown in Figure 2. The intensity and position of the diffraction peaks were identical, indicating that the four slag samples contained the same mineral phase, namely, spinel and dicalcium silicate. In other words, the mineral phase structures remained constant as the heating time increased. Thus, prolonging the heating time at 1450 °C had no adverse effect on the mineral phase structures of the CaO-SiO$_2$-MgO-Al$_2$O$_3$-Cr$_2$O$_3$-FeO system.

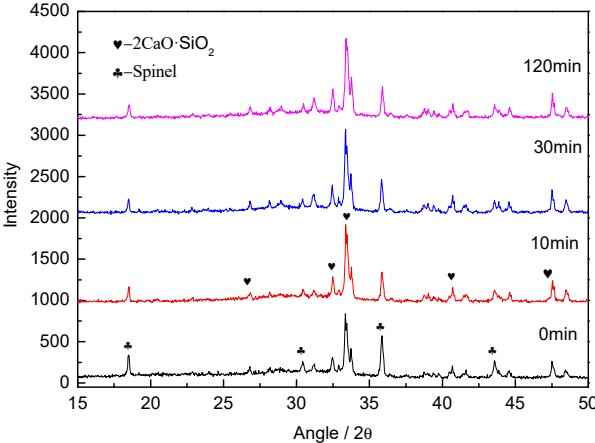

**Figure 2.** Effect of heat treatment time on the XRD diffraction pattern of the slag samples at 1450 °C.

The microstructures of the slag samples observed by SEM-EDS are shown in Figure 3. There were three kinds of mineral phases in all samples: (1) the white and regularly shaped phase is the spinel crystal, (2) the black striped phase is the $\alpha$-C$_2$S phase, and (3) the hoar phase is the glassy matrix. The proportion of spinel phase with respect to the other phases remained nearly stable for all of the executed treatments.

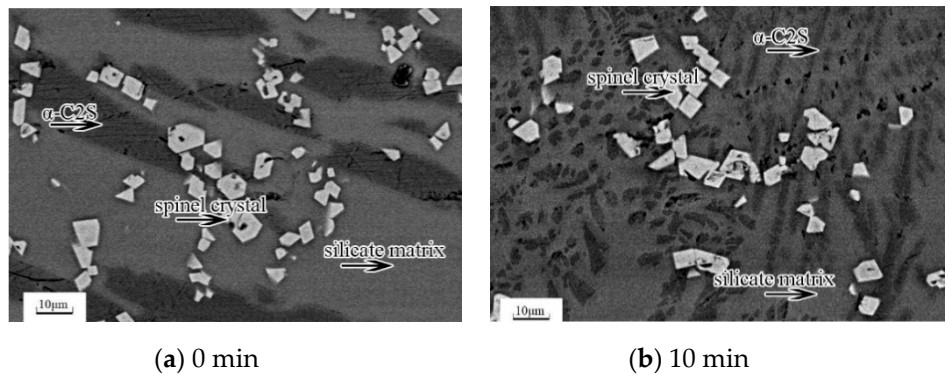

(**a**) 0 min          (**b**) 10 min

**Figure 3.** *Cont.*

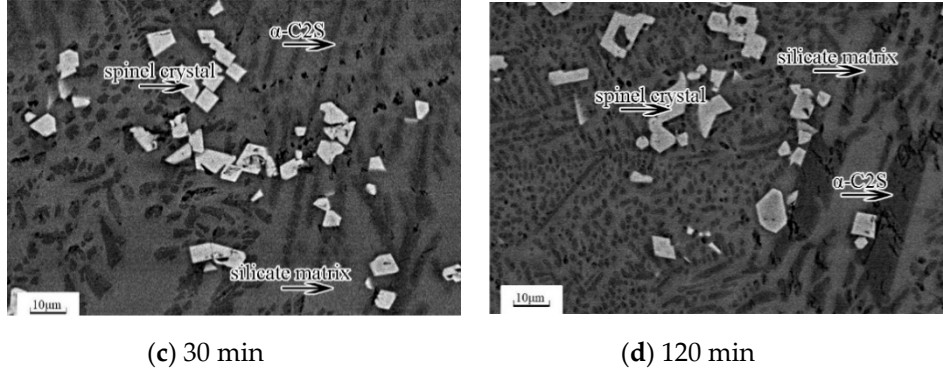

(**c**) 30 min            (**d**) 120 min

**Figure 3.** Effect of heating time on the microstructure of the slag samples at 1450 °C.

### 3.2. Behavior of Spinel Crystal in CaO-SiO₂-MgO-Al₂O₃-Cr₂O₃-FeO System

*3.2. Behavior of Spinel Crystal in $CaO$-$SiO_2$-$MgO$-$Al_2O_3$-$Cr_2O_3$-$FeO$ System*

The chemical compositions of spinel are shown in Table 2. When the heating time was prolonged from 0 min to 120 min, the calcium content decreased gradually from 1.38 at% to 0.68 at% and the chromium content increased from 16.55 at% to 22.78 at%. The content of calcium and iron showed a downward trend in spinel crystals. Nevertheless, the magnesium content showed an upward trend. As shown in Figure 4, the size of the spinel crystals increased from 9.42 μm to 10.73 μm when increasing the heating time from 0 min to 120 min, and the increase in the spinel size reached 13.91%.

**Table 2.** Chemical composition of the spinel crystals in Figure 4. Unit = atom%.

| Heating Time | Cr | O | Fe | Mg | Al | Ca |
|---|---|---|---|---|---|---|
| 0 min | 16.55 | 67.09 | 4.84 | 6.96 | 3.17 | 1.38 |
| 10 min | 20.56 | 60.91 | 5.18 | 8.28 | 3.76 | 1.32 |
| 30 min | 20.94 | 61.27 | 4.65 | 8.31 | 3.55 | 1.22 |
| 120 min | 22.78 | 60.22 | 4.25 | 8.44 | 3.61 | 0.68 |

Cao et al. [12] showed that the size of the spinel crystals increased from 10.8 μm to 24.8 μm during the cooling process from 1350 °C to 1250 °C and the increase reached 129% with respect to the initial size of the spinel crystals. The increase reached 108% (4.25 μm to 8.88 μm), with the silicon content increasing from 5 wt% to 15 wt% [13]. Moreover, with the addition of FeO increasing from 2 wt% to 20 wt%, the increase in size reached 46% [8]. Compared with the literature data, the increase in the spinel size was moderate in the case studied here.

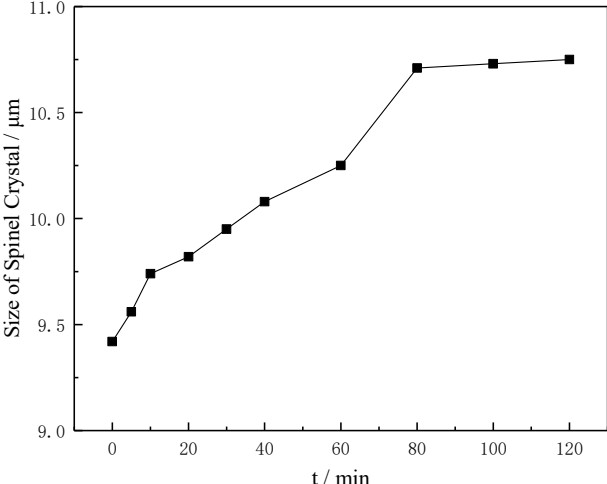

**Figure 4.** Effect of heat treatment time on the size of the spinel crystals at 1450 °C.

## 4. Discussion

### 4.1. Thermodynamic Considerations

The crystallization process of the molten slag is composed of nucleus formation and crystal growth, and is also determined by thermodynamic and kinetic factors. As shown in Figure 5, according to the calculation results of the solidification process through FactSage7.1, there are three phases including the liquid slag, spinel crystal, and $\alpha$-C$_2$S in the CaO-SiO$_2$-MgO-Al$_2$O$_3$-Cr$_2$O$_3$-FeO system at 1450 °C. Combined with the preparation of synthetic slag, the nucleus could not form in the liquid phase, which transforms directly into the glassy matrix, whereas spinel crystal and $\alpha$-C$_2$S were high-temperature precipitated phases in the CaO-SiO$_2$-MgO-Al$_2$O$_3$-Cr$_2$O$_3$-FeO system at 1450 °C. The heating time only affected the diffusion of the particles, and had little effect on mineral composition. These observations can explain why the mineral phases remained constant with the extension of heating time at 1450 °C.

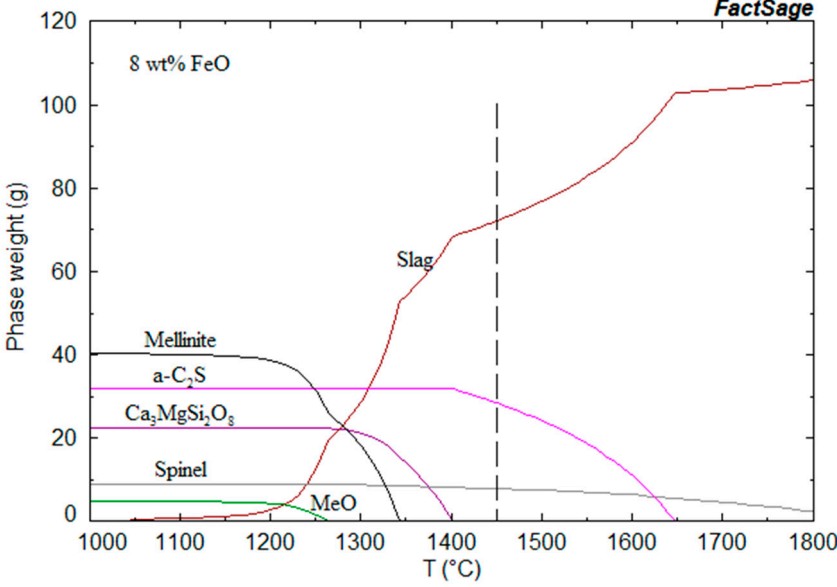

**Figure 5.** Theoretical analysis of the slag solidification process and selection of temperature.

Spinel crystal is the main mineral phase for improving chromium stability in the CaO-SiO$_2$-MgO-Al$_2$O$_3$-Cr$_2$O$_3$-FeO system [14–16]. Additionally, the reactions that form spinel crystals can occur between the MgO, Al$_2$O$_3$, Cr$_2$O$_3$, CaO, and FeO in the CaO-SiO$_2$-MgO-Al$_2$O$_3$-Cr$_2$O$_3$-FeO system. The reactions are shown in Equations (1)–(5).

$$(MgO) + (Cr_2O_3) = MgO \cdot Cr_2O_3(s) \tag{1}$$

$$(FeO) + (Cr_2O_3) = FeO \cdot Cr_2O_3(s) \tag{2}$$

$$(MgO) + (Al_2O_3) = MgO \cdot Al_2O_3(s) \tag{3}$$

$$(FeO) + (Al_2O_3) = FeO \cdot Al_2O_3(s) \tag{4}$$

$$(CaO) + (Cr_2O_3) = CaO \cdot Cr_2O_3(s) \tag{5}$$

MgCr$_2$O$_4$ formed by MgO and Cr$_2$O$_3$ is the most stable phase at high temperature [15]. During the heating period, iron and calcium elements that evolve into spinels through the isomorphic substitution are gradually replaced by magnesium. Consequently, the magnesium content increases, while the contents of iron and calcium decrease. Furthermore, chromium stability is affected by calcium content to a large extent [15]. The silicate microcrystals containing calcium are adsorbed into the spinel lattice during spinel growth due to the formation of a finite solid solution with

MgO·Cr$_2$O$_3$ [17]. Compared with other silicate mineral phases, dicalcium silicate is more soluble in water. The dissolution of chromium increases significantly as the content of dicalcium silicate in chromium spinels increases [17,18]. Thus, prolonging the heating time can reduce the content of dicalcium silicate in chromium spinels and improve their stability.

### 4.2. Precipitation and Growth of Spinel Crystals

The precipitation of spinel crystals includes nucleation and crystal growth. The slag system satisfies the basic requirements of supersaturation and supercooling for crystal nucleation and the essence of crystal nucleus growth is the transfer of atoms and other particles in liquid to the surface of the crystal nucleus. The composition of spinel crystals is obviously different from the chemical composition of slag and its growth rate depends on the long-range diffusion of solute atoms, which is essential to ensure the continuous growth of crystal [19,20]. For a slag system under constant temperature, the chemical reaction merely occurs between the free oxides such as (FeO), (Cr$_2$O$_3$), (MgO), (Al$_2$O$_3$), (CaO) based on slag molecular theory. The ability of molecules in slag to carry out chemical reactions is closely related to their activity, and the concentration of free oxides is the activity. Therefore, the reaction rate of Equations (1)–(5) are related to the concentrations of (FeO), (Cr$_2$O$_3$), (MgO), (Al$_2$O$_3$) and (CaO). As shown in Figure 6, various spinel crystals are precipitated when the slag with the concentration of solute atom $C_0$ is cooled to the temperature T. Furthermore, the concentration of solute atoms in the parent phase and precipitated phase at the phase interface is $C_M$ and $C_N$, respectively. During the *dt* time, the phase boundary pushes the distance of *dr* toward the parent phase, and the quantity of solute atoms needed for the volume increase of the precipitated phase is $(C_N - C_M)\,dr$. Moreover, this part of the solute is provided by the diffusion of solute atoms in the parent phase.

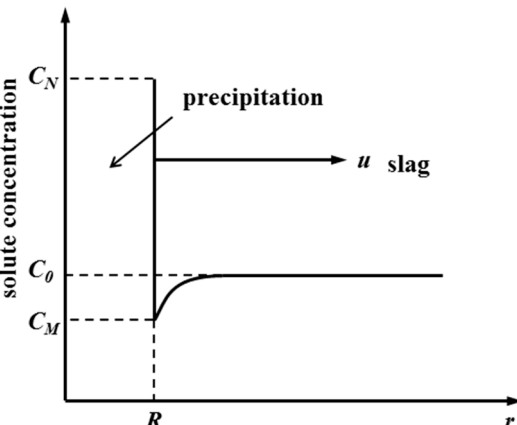

**Figure 6.** Concentration distribution of the solute atoms in slag during spinel crystal growth. $C_0$ represents the average concentration of solute atoms in the matrix, $C_M$ represents the concentration of solute atoms in the matrix side at the crystal interface, $C_N$ represents the concentration of solute atoms in the crystal side at the crystal interface, and *u* represents the growth rate of the crystal.

The growth rate of the spinel crystals can be reflected by Equations (6) and (7) [21]:

$$u = \frac{dr}{dt} = \frac{D}{c_N - c_M}\left(\frac{\partial c}{\partial r}\right)_{r=R} \tag{6}$$

$$D = \frac{k_B T}{3\pi d\mu} \tag{7}$$

where *d*, $\mu$, *T*, and $k_B$ refer to the particle diameter, melt viscosity, absolute temperature, and Boltzmann constant, respectively. $\left(\frac{\partial c}{\partial r}\right)_{r=R}$ refers to the concentration gradient of the particle near the phase interface.

It can be seen from the above equation that the growth rate of the precipitated phase crystal nucleus is directly proportional to the diffusion coefficient of the solute in the parent phase and the concentration gradient of the solute atom near the phase interface, while being inversely proportional to the difference in the equilibrium concentration between the two phases at the interface. For a system with a constant temperature, the diffusion coefficient D (cm$^2$/s, the order of magnitude is $10^{-5}$–$10^{-7}$ generally) is constant and is closely related to the viscosity in slag. In this way, the growth rate of spinel crystals mainly depends on the particle concentration difference ($C_N - C_M$) and concentration gradient ($(\frac{\partial c}{\partial r})_{r=R}$) at the interface of the spinel crystal. After the particle reaches the interface, it is rapidly consumed, meaning that the concentration difference of components near the phase interface is basically unchanged. For convenience of discussion, the ($C_N - C_M$) of components near the phase interface was considered approximately as a constant. Therefore, the effect of heating time on the growth rate of the spinel crystals mainly depends on the concentration gradient of the particles in the melt.

As shown in Figure 7, the theory of the spinel transition fraction X (X = process/final precipitation amount × 100) reached 88.37%, 92.25%, and 96.29% at 1450 °C, 1400 °C and 1350 °C, respectively. This reveals that the process of spinel crystallization is close to the terminal at the experimental temperature. In addition, during the solidification of the CaO-SiO$_2$-MgO-Al$_2$O$_3$-Cr$_2$O$_3$-FeO system, the contents of chromium in the residual liquid-slag and spinel crystals are as shown in Figure 8. At 1450 °C, the chromium content in the liquid phase was only 0.49%, which was far less than that in the spinel crystals. Obviously, the concentration gradient of particles was small due to the completion of the precipitation region of the spinel crystals and the limited content of chromium in the liquid phase. Moreover, the content of the residual liquid phase decreased with the completion of spinel crystal crystallization. The diffusion condition of the solute atoms becomes worse, which causes the diffusion coefficient *D* to become evidently low. Slow particle diffusion results in the prolongation of the heating time having little effect on the increase in the size of the spinel crystals. If the holding point is carried out at a higher temperature, the higher content of chromium in the liquid phase and the higher concentration gradient of the solute will increase, leading to increases in the growth rate of the spinel crystals and the size of spinels.

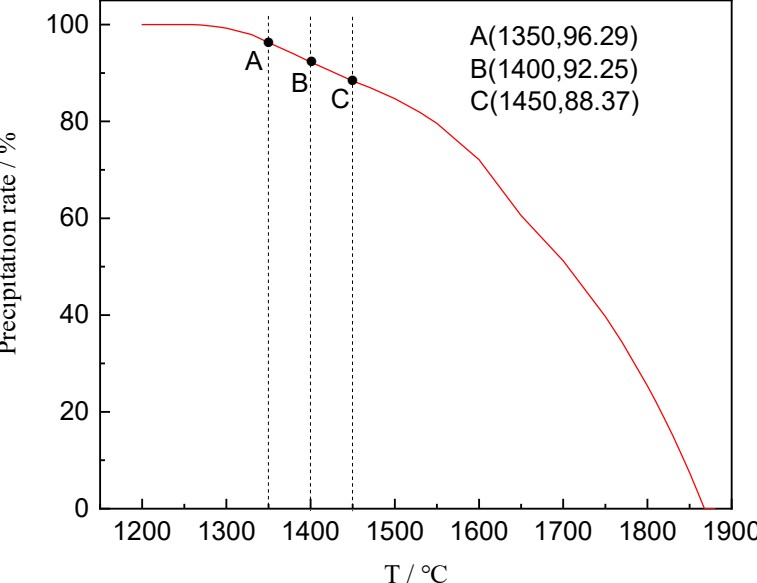

**Figure 7.** Theoretical transition fraction of a spinel crystal.

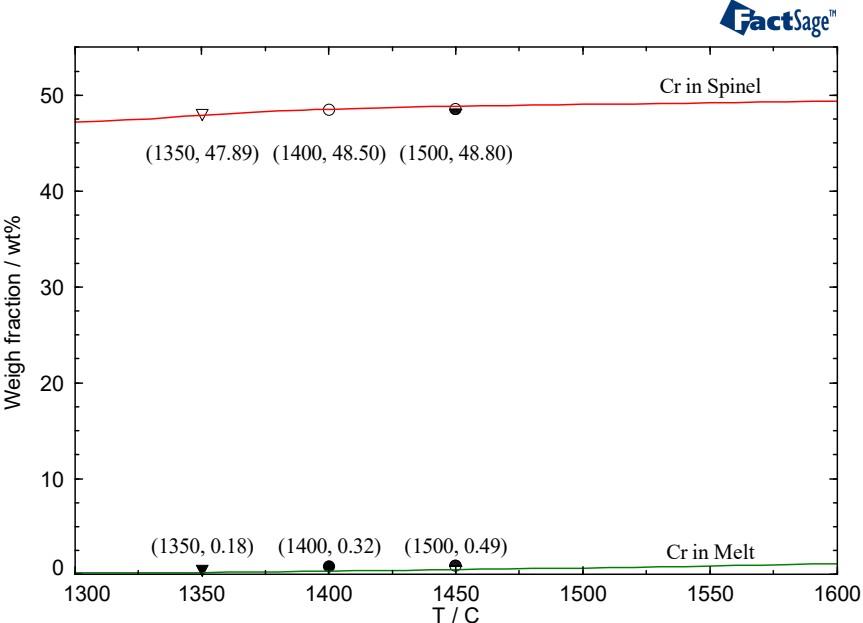

**Figure 8.** Content of chromium in the spinel crystal and liquid phase during slag solidification.

## 5. Conclusions

The precipitation behavior of spinel crystals in the $CaO-SiO_2-MgO-Al_2O_3-Cr_2O_3-FeO$ system was studied by means of SEM-EDS, XRD, and IPP in the laboratory. Based on this study, the following conclusions were made:

(1) When the heating time increased from 0 min to 120 min at 1450 °C, the species of mineral phases precipitated from the $CaO-SiO_2-MgO-Al_2O_3-Cr_2O_3-FeO$ system remained constant, and consisted of spinel and dicalcium silicate.

(2) The size of the spinel crystals increased from 9.42 to 10.73 μm with an increase in the heating time from 0 to 120 min. The increase in the spinel size reached 13.91%, which is considered moderate. The theoretical transformation fraction of the spinel reached 88.37% at 1450 °C. The crystallization process of the spinel occurred as a result of the low content of chromium and magnesium, forming the spinel in residual liquid, and the high viscosity of the solid-liquid mixture is a critical factor.

(3) The calcium content in the spinel crystal decreased gradually from 1.38 to 0.78 at%, while the chromium content increased from 16.55 to 22.78 at%. During the long-term heating, $MgCr_2O_4$ consisting of MgO and $Cr_2O_3$ was the most stable phase. The iron and calcium elements involved in the isomorphic substitution could be gradually replaced by magnesium in the spinel crystals.

**Author Contributions:** Y.Y. and Q.Z. helped to carry out the experimental work; Q.M. drafted the manuscript and conducted the experiments; J.L. analyzed the experimental data and modified and polished the draft.

**Funding:** The research was supported by the National Natural Science Foundation of China (No. 51404173), the Hubei Provincial Natural Science Foundation (No. 2016CFB579), the China Postdoctoral Science Foundation (No. 2014M562073), and the State Key Laboratory of Refractories and Metallurgy.

**Conflicts of Interest:** The authors declare no conflicts of interest.

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
