# Peer review of "Experimental Study on Precipitation Behavior of Spinels in Stainless Steel-Making Slag under Heating Treatment"

_processes, doi:10.3390/pr7080487_

Reviewer 1 Report

The article is well written and deals with the theme which is in the center of attention of many steelmakers.

In text are some minor inconsistencies marked in the attached file. 

Reviewer 2 Report

The manuscript Experimental Study on Precipitation Behavior of Spinel in Stainless Steel-making Slag under Heating Treatment written by Li et al. present interesting laboratory study concerning the issue of how to improve chromium stability in slag. I think this paper is written in a correct way, however some aspects need to be clarified in order to improve dispersal of “take-home-message” to the wider audience if the readers. In other words, the authors have to be more precise in some aspects so that not only reader familiar with the subject but yet people who intend to learn will understand the authors statement better. Some of these aspects I pointed out in.my detailed comments given below. To conclude, I have read this paper with a pleasure and I consider this data valuable for publishing however I recommend the Authors to revise this paper. Minor revision is reccoemnded.

The authors stated that: 

The stability of chromium in stainless steel slag has a positive correlation with the particle size please specify that you refer to spinel size rather than slag particle size, it is obvious for me but not necessary to the reader not familaiar with this subject

SEM-EDS and XRD were adopted please avoid abbreviations in the abstract

Introduction:

And the stainless steel slag – please do not begin the sentence with “And”

EAF slag and AOD slag- please explain abbbreviations in the brackets

it’s – please use unabbreviated form “it is” – please check it out throughout the whole manuscript

inhibit – inhibits

The raw materials included CaO, SiO2, MgO, Al2O3, Cr2O3 and FeC2O4·2H2O. Firstly, raw materials were weighed respectively and then mixed – please specify what amount was used for each

The proportion of spinel phase almost remains stable – Please rephrase eg. The proportion of spinel phase with respect to the other phases remains nearly stable for all executed treatments

the content of magnesium and iron shows a downward trend in spinel crystals – According to table the content of Mg increaseses so this statement is valid for Mg only, isn’t it?

Reference [11] – please refer to the authors names rather than to the “reference”

reached to 129%. – please rephrase reached 129% with repsect to the initial…

reached to 46% - reached 46%

Comparing with these references, the increase is moderate in this paper – please rephrase eg. 

Comparing with the literature data, the increase of spinel size is moderate in case studied here

The crystallization process of the molten slag is composed of nucleus formation and crystal growth, and IT IS also determined by thermodynamic and kinetic factors

Whereas, the spinel crystal and α-C2S are the high-temperature precipitated phaseS in

Thus, these can explain that the mineral phases REMAIN constant with the extension of heating time at 1450℃

Consequently, magnesium content increases, while the contents of iron and calcium decrease – YES this is confirmed by the Autohors  dataso please see my comment above

depends on the particle concentration difference and concentration gradient at the interface of spinel crystal – please explain in a more detailed manner particle concentration difference(?) and concentration of XX gradient (?)

Could the Authors please add one additional comment on Figure 7, regarding whether the temperature slightly lower eg. 1200 would be more suitable for complete crystallization? How would it affect composition? Is it possible to elaborate on this context?

The precipitation behavior of spinel crystals in CaO-SiO2-MgO-Al2O3-Cr2O3-FeO system was studied by means of SEM-EDS, XRD, and IPP in the laboratory, BASED ON THIS STUDY FOLLOWING CONCLUSIONS HAVE BEEN MADE: (rephrase as I proposed or in another way)

(1) mineral phase remain constant – please specify constant size, composition, volumetric proportion – I know what the Authors meant however I am concenred it will not be as much clear for potential reader not familaiar with subject

Technical comment: the authors contributions are not stated.
